# Spectral Heterogeneous Graph Convolutions via Positive Noncommutative Polynomials

## ABSTRACT

Heterogeneous Graph Neural Networks (HGNNs) have gained significant popularity in various heterogeneous graph learning tasks. However, most existing HGNNs rely on spatial domain-based methods to aggregate information, i.e., manually selected meta-paths or some heuristic modules, lacking theoretical guarantees. Furthermore, these methods cannot learn arbitrary valid heterogeneous graph filters within the spectral domain, which have limited expressiveness. To tackle these issues, we present a positive spectral heterogeneous graph convolution via positive noncommutative polynomials. Then, using this convolution, we propose PSHGCN, a novel heterogeneous graph convolutional network. PSHGCN offers a simple yet effective method for learning valid heterogeneous graph filters. Moreover, we demonstrate the rationale of PSHGCN in the graph optimization framework. We conducted an extensive experimental study to show that PSHGCN can learn diverse heterogeneous graph filters and outperform all baselines on open benchmarks. Notably, PSHGCN exhibits remarkable scalability, efficiently handling large real-world graphs comprising millions of nodes and edges. Our codes are available in the anonymous link: https://anonymous.4open.science/r/PSHGCN_Code-DFDC.

## CCS CONCEPTS

• **Mathematics of computing** → **Graph algorithms**; • **Computing methodologies** → **Neural networks**.

## KEYWORDS

Heterogenous Graph Neural Networks, Spectral Graph Convolutions, Positive Noncommutative Polynomials.

**ACM Reference Format:**

Anonymous Author(s). 2018. Spectral Heterogeneous Graph Convolutions via Positive Noncommutative Polynomials. In *Proceedings of Make sure to enter the correct conference title from your rights confirmation emai (Conference acronym 'XX)*. ACM, New York, NY, USA, 12 pages. https://doi.org/XXXXXXX.XXXXXXX

## 1 INTRODUCTION

In recent years, there has been a significant surge of interest in graph neural networks (GNNs) due to their remarkable performance in tackling diverse graph learning tasks, including but not limited to node classification [6, 15, 27], link prediction [3, 41, 45],

and graph property prediction [9, 17, 33]. While earlier versions of GNNs [15, 27] were primarily developed for homogeneous graphs, which consist of only one type of node and edge, real-world graphs typically comprise a diverse range of nodes and edges known as heterogeneous graphs. For instance, an academic graph may include multiple nodes such as "author," "paper," and "conference," as well as several edges such as "cite," "write," and "publish." Due to the extensive and diverse information in heterogeneous graphs, specialized models are necessary to analyze them effectively.

In response to the challenge of heterogeneity, numerous Heterogeneous Graph Neural Networks (HGNNs) have been proposed, achieving significant performance [1, 7, 28, 38]. The majority of these HGNNs depend on spatial domain-based message passing and attention modules for information propagation and aggregation. Following [35], we can broadly classify these HGNNs into two categories based on whether they use manually selected meta-paths or meta-path-free techniques to aggregate information.

Meta-path-based HGNNs [7, 28, 35, 40] begin by manually selecting or predefining specific meta-paths. Then, they perform message aggregation based on these meta-paths to obtain the final embedding. These message aggregation strategies encompass attention modules, Transformers, and various other techniques. In contrast, meta-path-free HGNNs [1, 18, 20, 31, 38, 39] create graph convolutions for heterogeneous graphs to propagate and aggregate messages. These convolutions are designed from the spatial domain, leveraging techniques like attention mechanisms and learnable weights to acquire node representations heuristically.

Although the above HGNNs have shown promising results in various heterogeneous graph learning tasks, they still have some significant limitations. First, the effectiveness of meta-path-based HGNNs relies on the manually selected meta-paths, resulting in poor theoretical guarantees. For example, SeHGNN [35] uses 41 meta-paths for feature aggregation on the ACM dataset, while HAN [28] uses only 6. This partly explains why SeHGNN outperforms HAN on this dataset. Furthermore, all these HGNNs design aggregation strategies or graph convolutions in the spatial domain heuristically, which cannot learn arbitrary graph filters like spectral-based GNNs [6, 10, 11, 30]. This results in limited expressiveness. For example, MHGCN [38] directly learns the summation weights of the adjacency matrix with only one type of edge, rendering it incapable of learning arbitrary filters. Additionally, these HGNNs acquire graph filters without any necessary constraints, making them challenging to learn, especially from a graph optimization perspective [10, 36, 43, 44], where graph filters should satisfy positive semidefinite constraints. In Section 6.4 of our experiments, we show the impact of the positive semidefinite constraint, and under equivalent conditions, models with this constraint perform better and exhibit reduced standard errors across multiple runs.

To address these issues, we first introduce the concept of spectral heterogeneous graph convolution, which is a straightforward and

intuitive extension of spectral graph convolution. Building upon this, we present a positive spectral heterogeneous graph convolution, leveraging positive noncommutative polynomials to ensure that the acquired graph filters maintain positive semidefiniteness. Using this convolutional approach, we propose a novel heterogeneous graph convolutional network named PSHGCN. PSHGCN offers a simple yet highly effective method for learning heterogeneous graph filters. Moreover, we analyze the rationale behind PSHGCN from the perspective of graph optimization. Our analysis shows that PSHGCN has the theoretical capacity to express a wide array of valid heterogeneous graph filters. Finally, we conduct an extensive experiment, demonstrating that PSHGCN excels in tasks such as node classification and link prediction. This underscores PSHGCN's ability to learn heterogeneous graph filters adeptly. Notably, PSHGCN exhibits remarkable scalability, efficiently handling large real-world heterogeneous graphs comprising millions of nodes and edges. We summarize the contributions of this paper as follows:

(1) We propose PSHGCN, a heterogeneous graph convolutional network that uses positive spectral heterogeneous graph convolution to learn valid heterogeneous graph filters.
(2) We present a generalized heterogeneous graph optimization framework and demonstrate the rationale of our PSHGCN from this framework.
(3) Thorough experiments demonstrate that PSHGCN achieves superior performance in tasks such as node classification and link prediction, and has desirable scalability.

## 2 RELATED WORK

**Graph Neural Networks (GNNs)** are machine-learning techniques designed specifically for graph data. These methods aim to find a low-dimensional vector representation for each node, enabling efficient processing for various network mining tasks. GNNs can be broadly classified into two categories: spatial-based and spectral-based approaches [32]. Spatial-based GNNs directly propagate and aggregate information in the spatial domain. From this viewpoint, GCN [15] can be interpreted as aggregating one-hop neighbor information in each layer. GAT [27] leverages attention mechanisms to learn aggregation weights.

Spectral-based GNNs utilize spectral graph convolutions/filters designed in the spectral domain. ChebNet [6] employs Chebyshev polynomials to approximate filters. GCN [15] simplifies the Chebyshev filter by utilizing a first-order approximation. APPNP [16] uses Personalized PageRank (PPR) to determine the filter weights. GPR-GNN [5] learns polynomial filters by employing gradient descent on the polynomial weights. BernNet [10] utilizes the Bernstein basis for approximating graph convolutions, enabling the learning of arbitrary graph filters. JacobiConv [30] and ChebNetII [11] use Jacobi polynomials and Chebyshev interpolation, respectively, to learn filters. OptBasisGNN [8] first computes the optimal polynomial bases and then uses them to learn filters. However, all these methods are designed for homogeneous graphs and do not perform optimally on heterogeneous graphs.

**Heterogeneous Graph Neural Networks (HGNNs)** are explicitly developed to address the challenges posed by heterogeneous graphs. HGNNs can be broadly categorized into meta-path-based

and meta-path-free HGNNs [35]. Meta-path-based HGNNs propagate and aggregate neighbor features using selected meta-paths. HAN [28] uses a hierarchical attention mechanism with multiple meta-paths for aggregating node features and semantic information. HetGNN [40] employs random walks to generate node neighbors and aggregates their features. MAGNN [7] encodes information from manually selected meta-paths instead of just focusing on endpoints. SeHGNN [35] utilizes predetermined meta-paths for neighbor aggregations and applies a transformer-based approach.

Meta-path-free HGNNs propagate and aggregate messages from neighboring nodes in a manner similar to GNNs, without requiring a selected meta-path. RGCN [20] extends GCN [15] to heterogeneous graphs with edge type-specific graph convolutions. GTN [39] utilizes soft sub-graph selection and matrix multiplication to generate meta-path neighbor graphs. SimpleHGN [18] incorporates a multi-layer GAT network with attention based on node features and learnable edge-type embeddings. MHGCN [38] directly learns the summation weights and employs GCN's convolution for feature aggregation. EMRGNN [31] and HALO [1] propose optimization objectives tailored for heterogeneous graphs and design architectures by solving these optimization problems. HINormer [19] uses the local structure encoder and the relation encoder, with graph Transformer to learn node embeddings. MGNN [2] uses noncommutative polynomials to create graph convolutions for multigraphs in the spectral domain. All of these HGNNs are designed within the spatial domain, except for MGNN. However, MGNN mainly focuses on multigraphs and has no constraints for learned filters. The lack of robust theoretical guarantees and expressiveness within spatial-based HGNNs, coupled with the limited exploration of spectral-based HGNNs, motivates us to propose the PSHGCN.

## 3 PRELIMINARIES

### 3.1 Spectral Graph Convolution

Recent studies suggest that many popular spectral-based GNNs utilize a polynomial of the Laplacian matrix to approximate spectral graph convolutions [6, 10, 11, 15, 30]. In particular, we denote an undirected homogeneous graph with node set $V$ and edge set $E$ as $G(V, E)$, whose adjacency matrix is $\mathbf{A}$. Let $\mathbf{L} = \mathbf{I} - \hat{\mathbf{A}} = \mathbf{I} - \mathbf{D}^{-1/2}\mathbf{A}\mathbf{D}^{-1/2}$ denote the normalized Laplacian matrix, where $\hat{\mathbf{A}} = \mathbf{D}^{-1/2}\mathbf{A}\mathbf{D}^{-1/2}$ denotes the normalized adjacency matrix and $\mathbf{D}$ is the diagonal degree matrix of $\mathbf{A}$, i.e., $\mathbf{D}[i, i] = \sum_j \mathbf{A}[i, j]$. We use $\mathbf{L} = \mathbf{U}\boldsymbol{\Lambda}\mathbf{U}^\top$ to represent the eigendecomposition of $\mathbf{L}$, where $\mathbf{U}$ denotes the matrix of eigenvectors and $\boldsymbol{\Lambda} = \text{diag}[\lambda_1, ..., \lambda_{|V|}]$ is the diagonal matrix of eigenvalues. Given a graph signal vector $\mathbf{x} \in \mathbb{R}^{|V|}$, the spectral graph convolution is defined as

$$\mathbf{y} = h(\mathbf{L})\mathbf{x} = \mathbf{U}h(\boldsymbol{\Lambda})\mathbf{U}^\top\mathbf{x} = \mathbf{U}\text{diag}\left[h(\lambda_1), ..., h(\lambda_{|V|})\right]\mathbf{U}^\top\mathbf{x}. \quad (1)$$

The function $h(\mathbf{L})$ (or, equivalently, $h(\lambda)$) is the **spectral graph filter** and $\mathbf{y}$ denotes the output of graph convolution. To learn filters while avoiding the expansive eigendecomposition, existing methods use polynomials to approximate $h(\mathbf{L})$ [6, 11].

$$\mathbf{y} = h(\mathbf{L})\mathbf{x} \approx \sum_{k=0}^{K} w_k \mathbf{L}^k \mathbf{x}, \quad (2)$$

where $w_k$ are the polynomial filter weights. We can obtain different filters by setting or learning the weights $w_k$.

## 3.2 Graph Optimization Framework

We can obtain the spectral graph convolution through the lens of a classical graph optimization problem [10, 44].

$$\min_{\mathbf{y}} f(\mathbf{y}) = (1 - \alpha)\mathbf{y}^\top \gamma(\mathbf{L})\mathbf{y} + \alpha \parallel \mathbf{y} - \mathbf{x} \parallel_2^2, \quad (3)$$

where the first term is a smooth operation of the signals based on the graph structure and $\gamma(\cdot)$ is an energy function [22]. The second term is a regularization that maintains the original signals. The parameter $\alpha \in (0, 1)$ is a trade-off parameter. We can get the closed-form solution of the problem (3) by setting $\frac{\partial f(\mathbf{y})}{\partial \mathbf{y}} = 0$.

$$\mathbf{y} = h(\mathbf{L})\mathbf{x} = \alpha(\alpha\mathbf{I} + (1 - \alpha)\gamma(\mathbf{L}))^{-1}\mathbf{x}. \quad (4)$$

Equation (4) can express some specific convolution by setting different functions $\gamma(\mathbf{L})$ [10, 44]. For example, if we set $\gamma(\mathbf{L}) = \mathbf{L}$, then $\mathbf{y} = \alpha(\alpha\mathbf{I} + (1 - \alpha)\mathbf{L})^{-1}\mathbf{x} = \alpha(\mathbf{I} - (1 - \alpha)\hat{\mathbf{A}})^{-1}\mathbf{x}$, corresponding the graph convolution of PPNP/APPNP [16]. We can obtain more existing GNNs' convolutions using Equation (4). For the details, please refer to papers [10, 44].

Importantly, in Equation (3), the output of function $\gamma(\mathbf{L})$ has to be positive semidefinite. If $\gamma(\mathbf{L})$ fail to satisfy this condition, the optimization function $f(\mathbf{y})$ becomes non-convex, and the solution to $\frac{\partial f(\mathbf{y})}{\partial \mathbf{y}} = 0$ may lead to a saddle point. When $\gamma(\mathbf{L})$ is positive semidefinite, we can derive that the spectral graph filter $h(\mathbf{L}) = \alpha(\alpha\mathbf{I} + (1 - \alpha)\gamma(\mathbf{L}))^{-1}$ is positive semidefinite, i.e., $h(\lambda) \geq 0$ [10]. Therefore, based on the graph optimization framework, a spectral graph filter $h(\mathbf{L})$ should be **positive semidefinite**.

## 3.3 Heterogeneous Graph

A heterogeneous graph [25] is defined as $G = (V, E, \phi, \psi)$, where $V$ is the set of nodes and $E$ is the set of edges. Let $n = |V|$ denote the number of nodes. Each node $v \in V$ is attached with a node type $\phi(v)$ and each edge $e \in E$ is attached with an edge type $\psi(e)$. We use $\mathcal{T}_v = \{\phi(v) : \forall v \in V\}$ to denote the set of possible node types and $\mathcal{T}_e = \{\psi(e) : \forall e \in E\}$ to denote the set of possible edge types. When $|\mathcal{T}_v| = |\mathcal{T}_e| = 1$, the graph becomes an ordinary homogeneous graph. For convenience, we use $R = |\mathcal{T}_e|$ to denote the number of edge types.

For a heterogeneous graph $G$, we denote the sub-graph generated by differentiating the types of edges between all nodes as $\{G_r | r = 1, 2, \ldots, R\}$. Each $G_r$ includes $n$ nodes but only contains one type of edge. Let $\mathbf{A}_r$ denote the adjacency matrix of the sub-graph $G_r$, where $\mathbf{A}_r[i, j]$ is non-zero if there exists an $r$-th type edge from $i$ to $j$. Notably, in the general case of heterogeneous graphs, the sub-graph $G_r$ is a directed graph. Hence, we use $\mathbf{D}_r$ to represent the diagonal out-degree matrix of $\mathbf{A}_r$, i.e., $\mathbf{D}_r[i, i] = \sum_j^n \mathbf{A}_r[i, j]$. We use $\hat{\mathbf{A}}_r = \mathbf{D}_r^{-1}\mathbf{A}_r$ to denote the normalized adjacency matrix and use $\mathbf{L}_r = \mathbf{I} - \hat{\mathbf{A}}_r$ to denote the normalized Laplacian matrix. For brevity, we assume that all nodes possess the same dimensional features and denote the node features as $\mathbf{X} \in \mathbb{R}^{n \times d}$, where $d$ represents the dimensionality of the node features.

## 4 THE PROPOSED METHOD: PSHGCN

In this section, we will begin by introducing the concept of spectral heterogeneous graph convolution. Subsequently, we will propose a positive spectral heterogeneous graph convolution, ensuring its

**Table 1: Existing HGNNs that attempt to design the spectral heterogeneous graph convolution.**

| Method | Shift $\mathbf{P}_r$ | Graph Convolution |
|---|---|---|
| GTN [39] | $\mathbf{A}_r$ | $\mathbf{D}^{-1} \prod_{k=0}^{K} \sum_{r=0}^{R} \alpha_r^{(k)} \mathbf{A}_r \mathbf{x}$ |
| EMRGNN [31] | $\tilde{\mathbf{A}}_r$ | $\sum_{k=0}^{K} \alpha(1 - \alpha)^k \left( \sum_{r=1}^{R} \mu_r \tilde{\mathbf{A}}_r \right)^k \mathbf{x}$ |
| MHGCN [38] | $\mathbf{A}_r$ | $\left( \sum_{r=1}^{R} \beta_r \mathbf{A}_r \right)^K \mathbf{x}$ |

positive semidefinite nature. Finally, we will provide a comprehensive overview of the implementation of the Positive Spectral Heterogeneous Graph Convolutional Network (PSHGCN).

## 4.1 Spectral Heterogeneous Graph Convolution

Expanding spectral graph convolution, i.e., Equation (2), to heterogeneous graphs is a straightforward and intuitive process. MGNN [2], in this context, has introduced a method for defining graph convolution on multigraphs through the utilization of noncommutative polynomials. This approach can be readily applied to heterogeneous graphs. Specifically, We use $\mathbf{P}_r$ to denote either the adjacency matrix $\hat{\mathbf{A}}_r$ or the Laplacian matrix $\mathbf{L}_r$ of sub-graph $G_r$. This $\mathbf{P}_r$ is commonly recognized as the shift operator in graph signal processing [23].

DEFINITION 4.1. (Spectral Heterogeneous Graph Convolution). *Consider a heterogeneous graph $G = (V, E, \phi, \psi)$ with shift operators $\{\mathbf{P}_r\}_{r=1}^{R}$. A spectral heterogeneous graph convolution of a graph signal $\mathbf{x} \in \mathbb{R}^n$ on $G$ is defined as $h(\mathbf{P}_1, \mathbf{P}_2, \ldots, \mathbf{P}_R)\mathbf{x}$, where $h$ denotes a noncommutative polynomial function that takes the shift operators $\{\mathbf{P}_r\}_{r=1}^{R}$ as independent variables.*

Here, we call $h(\mathbf{P}_1, \mathbf{P}_2, \ldots, \mathbf{P}_R)$ the heterogeneous graph filter and formalize it as $w_0\mathbf{I} + \sum_{k=1}^{K} \sum_{r_1, r_2, \ldots, r_k} w_{r_1, r_2, \ldots, r_k} (\mathbf{P}_{r_1}\mathbf{P}_{r_2} \ldots \mathbf{P}_{r_k})$, where $w_{r_1, r_2, \ldots, r_k} \in \mathbb{R}$ denote the polynomial coefficients, $K \in \mathbb{Z}^+$ is the order of the polynomial, and $r_i \in \{1, 2, \ldots, R\}$ for each $i \in \{1, 2, \ldots, k\}$. For example, a 2-order polynomial filter $h$ with two variables can be denoted as $h(\mathbf{P}_1, \mathbf{P}_2) = w_0\mathbf{I} + w_1\mathbf{P}_1 + w_2\mathbf{P}_2 + w_{1,1}\mathbf{P}_1\mathbf{P}_1 + w_{1,2}\mathbf{P}_1\mathbf{P}_2 + w_{2,1}\mathbf{P}_2\mathbf{P}_1 + w_{2,2}\mathbf{P}_2\mathbf{P}_2$.

Some existing HGNNs can be perceived as attempts to design the spectral heterogeneous graph convolution. We show the details in Table 1. Specifically, GTN [39] introduces a Graph Transformer layer to perform the graph convolution, which can be represented as $\mathbf{D}^{-1} \prod_{k=0}^{K} \sum_{r=0}^{R} \alpha_r^{(k)} \mathbf{A}_r \mathbf{x}$. In this expression, $\alpha_r^{(k)}$ are learnable weights, $\mathbf{D}$ is the degree matrix for normalization, and $\mathbf{A}_0$ is the identity matrix. EMRGNN [31] utilizes the multi-relational Personalized PageRank [16] to design the graph convolutions, which can be expressed as $\sum_{k=0}^{K} \alpha(1 - \alpha)^k \left( \sum_{r=1}^{R} \mu_r \tilde{\mathbf{A}}_r \right)^k \mathbf{x}$. Here, $\tilde{\mathbf{A}}_r$ is the normalized adjacency matrix with self-loops. EMRGNN approximates a graph filter by learning the weight $\mu_r$ and setting $\alpha$ as a non-negative trade-off parameter. MHGCN [38] applies a straightforward approach by aggregating the adjacency matrix $\mathbf{A}_r$ with learnable weights $\beta_r$, and uses $K$ GCN-layers to achieve the graph convolution. This can be expressed as $\left( \sum_{r=1}^{R} \beta_r \mathbf{A}_r \right)^K \mathbf{x}$.

We can observe that the above methods try to approximate the heterogeneous graph filter by learning different weights. However, these methods constitute specific instances of the graph filter

Figure 1: An illustration of the proposed PSHGCN.

$h(\mathbf{P}_1, \mathbf{P}_2, \ldots, \mathbf{P}_R)$, i.e., they cannot be equivalent to this noncommutative polynomial $h$, which limits their expressiveness. In fact, the polynomial graph filter $h$ possesses the capacity to approximate arbitrary filter functions, given that the order $K$ is sufficiently high.

## 4.2 Positive Spectral Heterogeneous Graph Convolution

Although employing the spectral heterogeneous graph convolution, as defined in Definition 4.1, seems promising, it does not inherently guarantee the learned graph filters are positive semidefinite. As discussed in Section 3.2, the valid spectral graph filter on homogeneous graphs should satisfy the positive semidefinite constraint. Furthermore, we will provide both theoretical and empirical evidence to support the assertion that graph filters on heterogeneous graphs should also conform to the positive semidefinite constraint in forthcoming sections.

To ensure the learned graph filters are positive semidefinite, spectral-based GNNs on homogeneous graphs have employed various techniques, such as Bernstein Approximation [10] and Polynomial Interpolation [11]. However, directly extending these methods to the heterogeneous filter $h(\mathbf{P}_1, \mathbf{P}_2, \ldots, \mathbf{P}_R)$ is infeasible since the shift operators $\{\mathbf{P}_r\}_{r=1}^R$ are noncommucative and share different eigenspace. Consequently, ensuring that heterogeneous filter $h$ meets the positive semidefinite constraint becomes a **nontrivial and challenging** problem. To address this problem, we propose to use the positive noncommutative polynomials [12], characterized by a Sum of Squares form, to redefine the spectral heterogeneous graph convolution.

DEFINITION 4.2. (Sum of Squares). *A noncommutative polynomial* $h(\mathbf{P}_1, \mathbf{P}_2, \ldots, \mathbf{P}_R)$ *is a Sum of Squares if it satisfies* $h(\mathbf{P}_1, \mathbf{P}_2, \ldots, \mathbf{P}_R) = \sum_i g_i(\mathbf{P}_1, \mathbf{P}_2, \ldots, \mathbf{P}_R)^\top g_i(\mathbf{P}_1, \mathbf{P}_2, \ldots, \mathbf{P}_R)$, *where each* $g_i$ *is an arbitrary polynomial and* $g_i^\top$ *denotes its transpose.*

If a noncommutative polynomial conforms to the Sum of Squares form, it must exhibit positive semidefinite properties, and the opposite also holds. Specifically, we have the following theorem.

THEOREM 4.1. [12] *Let* $h(\mathbf{P}_1, \mathbf{P}_2, \ldots, \mathbf{P}_R)$ *denote a noncommutative polynomial. If* $h(\mathbf{P}_1, \mathbf{P}_2, \ldots, \mathbf{P}_R)$ *conforms to the Sum of Squares form, then* $h(\mathbf{P}_1, \mathbf{P}_2, \ldots, \mathbf{P}_R)$ *is positive semidefinite. Conversely, If the* $h(\mathbf{P}_1, \mathbf{P}_2, \ldots, \mathbf{P}_R)$ *is positive semidefinite, then* $h(\mathbf{P}_1, \mathbf{P}_2, \ldots, \mathbf{P}_R)$ *meets the Sum of Squares form.*

Theorem 4.1 shows the necessity of using a Sum of Squares to ensure that $h(\mathbf{P}_1, \mathbf{P}_2, \ldots, \mathbf{P}_R)$ is positive semidefinite. Based on this, we propose the positive spectral heterogeneous graph convolution.

DEFINITION 4.3. (Positive Spectral Heterogeneous Graph Convolution). *Consider a heterogeneous graph* $G = (V, E, \phi, \psi)$ *with shift operators* $\{\mathbf{P}_r\}_{r=1}^R$. *A positive spectral heterogeneous graph convolution of a signal* $\mathbf{x}$ *is defined as* $\sum_i g_i(\mathbf{P}_1, \mathbf{P}_2, \ldots, \mathbf{P}_R)^\top g_i(\mathbf{P}_1, \mathbf{P}_2, \ldots, \mathbf{P}_R)\mathbf{x}$, *where each* $g_i$ *denotes an arbitrary polynomial and* $g_i^\top$ *is its transpose.*

## 4.3 Implementation of PSHGCN

According to Definition 4.3, it is possible to acquire arbitrary filters that satisfy positive semidefinite constraints by learning various polynomial functions $g_i$. However, learning multiple functions $g_i$ is challenging in practice. Therefore, we simplify the Sum of Squares form by utilizing a single polynomial $g$. It is essentially an arbitrary monomial noncommutative polynomial. Remarkably, despite focusing solely on learning a single polynomial function, the experiments in Section 6 demonstrate that this approach shows excellent performance. Additionally, in the implementation of PSHGCN, we opt for $\mathbf{P}_r = \hat{\mathbf{A}}_r$. As a result, the convolution of PSHGCN is

$$\mathbf{y} = g(\hat{\mathbf{A}}_1, \hat{\mathbf{A}}_2, \ldots, \hat{\mathbf{A}}_R)^\top g(\hat{\mathbf{A}}_1, \hat{\mathbf{A}}_2, \ldots, \hat{\mathbf{A}}_R)\mathbf{x}, \quad (5)$$

where $\mathbf{x} \in \mathbb{R}^n$ represents a graph signal and we treat it as a column of the node features $\mathbf{X}$. As illustrated in Figure 1, PSHGCN acquires a heterogeneous graph filter by learning the polynomial $g$.

In practice, many heterogeneous graphs exhibit varying dimensional features for different node types. To address this, we employ multiple Multi-layer Perceptrons (MLPs) for feature projection, aligning them into a common dimensional space, a strategy commonly employed by many existing HGNNs [18, 31, 35]. Subsequently, we apply the graph convolution as specified in Equation (5), and $g$ is a noncommutative polynomial. That is $g(\hat{\mathbf{A}}_1, \hat{\mathbf{A}}_2, \ldots, \hat{\mathbf{A}}_R) = w_0\mathbf{I} + \sum_{k=1}^K \sum w_{r_1, r_2, \ldots, r_k} \left( \hat{\mathbf{A}}_{r_1} \hat{\mathbf{A}}_{r_2} \cdots \hat{\mathbf{A}}_{r_k} \right)$, where $w_0$ and $w_{r_1, r_2, \ldots, r_k}$ are learnable coefficients. Finally, we use an MLP for downstream tasks. More precisely, the model structure of PSHGCN can be formulated as

$$\mathbf{H} = \text{MLP}_{in}(\mathbf{X}),$$
$$\mathbf{Y} = g(\hat{\mathbf{A}}_1, \hat{\mathbf{A}}_2, \ldots, \hat{\mathbf{A}}_R)^\top g(\hat{\mathbf{A}}_1, \hat{\mathbf{A}}_2, \ldots, \hat{\mathbf{A}}_R)\mathbf{H}, \quad (6)$$
$$\mathbf{Z} = \text{MLP}_{out}(\mathbf{Y}).$$

Notably, the original node features may span diverse dimensions, resulting in the existence of multiple $\text{MLP}_{in}$ modules. For the sake of clarity and simplicity, we opt for a simplified form. For a more detailed description, please refer to Algorithm 1 in Appendix C.

**Decoupled PSHGCN.** Similar to many spectral-based GNNs [8, 11, 16], our PSHGCN can be extended to large-scale graphs by decoupling the feature transformation and propagation processes. In particular, we first calculate and store $\hat{\mathbf{A}}_{r_1}\hat{\mathbf{A}}_{r_2}\cdots\hat{\mathbf{A}}_{r_k}\mathbf{X}$ for the original feature $\mathbf{X}$ in the preprocessing. Then we perform graph convolution operations. We have the following special process.

$$\mathbf{Y} = c_0\mathbf{X} + \sum_{k=1}^{K}\sum c_{r_1,r_2,\dots,r_k}\hat{\mathbf{A}}_{r_1}\hat{\mathbf{A}}_{r_2}\cdots\hat{\mathbf{A}}_{r_k}\mathbf{X},$$
$$\mathbf{Z} = \text{MLP}_{out}(\mathbf{Y}). \tag{7}$$

Here, $c_{r_1,r_2,\dots,r_k}$ denote the coefficients of the $\hat{\mathbf{A}}_{r_1}\hat{\mathbf{A}}_{r_2}\cdots\hat{\mathbf{A}}_{r_k}$ term in the expansion of $g(\hat{\mathbf{A}}_1,\hat{\mathbf{A}}_2,\dots,\hat{\mathbf{A}}_R)^{\top}g(\hat{\mathbf{A}}_1,\hat{\mathbf{A}}_2,\dots,\hat{\mathbf{A}}_R)$. The precomputed $\hat{\mathbf{A}}_{r_1}\hat{\mathbf{A}}_{r_2}\cdots\hat{\mathbf{A}}_{r_k}\mathbf{X}$ allows us to train PSHGCN in a mini-batch manner. For more details, please check Algorithm 2 in Appendix C. In our experiments, we assess the scalability of PSHGCN on ogbn-mag and find that PSHGCN achieves a new SOTA result.

## 5 MODEL ANALYSIS

In this section, we will demonstrate the necessity of the positive semidefinite constraint from the graph optimization perspective for heterogeneous graph filters. Meanwhile, we will elaborate on the rationale behind using PSHGCN and the theoretical guarantees of its effectiveness. Finally, we will analyze the complexity.

### 5.1 Understanding PSHGCN from the Graph Optimization Perspective

**Generalized Heterogeneous Graph Optimization Framework.** The utilization of graph optimization in designing GNNs for homogeneous graphs has been extensively explored and has led to remarkable performance [10, 36, 43, 44]. However, there have been limited efforts to extend the graph optimization framework to heterogeneous graphs. Based on the graph optimization framework for homogeneous graphs discussed in Section 3.2, we introduce a generalized heterogeneous graph optimization problem

$$\min_{\mathbf{y}} f(\mathbf{y}) = (1-\alpha)\mathbf{y}^{\top}\gamma(\mathbf{P}_1,\mathbf{P}_2,\dots,\mathbf{P}_R)\mathbf{y} + \alpha \parallel \mathbf{y} - \mathbf{x} \parallel_2^2, \tag{8}$$

where $\alpha \in (0,1)$ is a trade-off parameter, $\mathbf{y}$ denotes the resulting representation of the input signal $\mathbf{x}$, and $\gamma(\mathbf{P}_1,\mathbf{P}_2,\dots,\mathbf{P}_R)$ is an energy function determining the rate of propagation [22]. Generally, $\gamma(\cdot)$ takes the shift operators $\{\mathbf{P}_r\}_{r=1}^{R}$ as inputs and produces a real $n \times n$ matrix. Similar to Equation (3), we require that $\gamma(\mathbf{P}_1,\mathbf{P}_2,\dots,\mathbf{P}_R)$ must be positive semidefinite, so that the optimization problem (8) has a closed-form solution. By setting the derivative $\frac{\partial f(\mathbf{y})}{\partial \mathbf{y}} = 0$, we can obtain this solution as

$$\mathbf{y} = \alpha\left[\alpha\mathbf{I} + (1-\alpha)\gamma(\mathbf{P}_1,\mathbf{P}_2,\dots,\mathbf{P}_R)\right]^{-1}\mathbf{x}. \tag{9}$$

We can set up specific $\gamma$ functions to get some of the existing HGNNs within this generalized graph optimization framework. For example, if we set $\mathbf{P}_r = \tilde{\mathbf{L}}_r$, where $\tilde{\mathbf{L}}_r$ is the normalized Laplacian matrix with self-loops, and $\gamma(\tilde{\mathbf{L}}_1,\tilde{\mathbf{L}}_2,\dots,\tilde{\mathbf{L}}_R) = \sum_{r=1}^{R}\mu_r\tilde{\mathbf{L}}_r$ subject to $\sum_{r=1}^{R}\mu_r = 1$ and $\mu_r \geq 0$, then we can get the solution

$\mathbf{y} = \alpha\left(\mathbf{I} - (1-\alpha)\sum_{r=1}^{R}\mu_r\tilde{\mathbf{A}}_r\right)^{-1}\mathbf{x}$, which is the heterogeneous graph convolution used in EMRGNN [31].

**Positive semidefinite constraint.** We can observe that the $\alpha\left[\alpha\mathbf{I} + (1-\alpha)\gamma(\mathbf{P}_1,\mathbf{P}_2,\dots,\mathbf{P}_R)\right]^{-1}$ in Equation (9) denotes the heterogeneous graph filter, and the $h(\mathbf{P}_1,\mathbf{P}_2,\dots,\mathbf{P}_R)$ defined in Definition 4.1 is its polynomial approximation. This is consistent with the concepts discussed on homogeneous graphs in Section 3.1. Within this graph optimization framework, we can deduce that the heterogeneous graph filter has to satisfy a positive semidefinite constraint. Specifically, we have the following lemma, the proof of which can be found in Appendix B.

LEMMA 5.1. *Consider an arbitrary function $\gamma(\mathbf{P}_1,\mathbf{P}_2,\dots,\mathbf{P}_R)$ that produces a real positive semidefinite $n \times n$ matrix and let $\alpha$ be in the interval $(0,1)$. Then the matrix $\alpha\left[\alpha\mathbf{I} + (1-\alpha)\gamma(\mathbf{P}_1,\mathbf{P}_2,\dots,\mathbf{P}_R)\right]^{-1}$ is also a real positive semidefinite matrix.*

A heterogeneous graph filter must satisfy the requirement of a positive semidefinite constraint. This fact motivates us to introduce the positive spectral heterogeneous graph convolution and the PSHGCN based on it. According to Theorem 4.1, the Sum of Squares form is a necessary and sufficient condition for ensuring the positive semidefinite constraint of heterogeneous graph filter $h(\mathbf{P}_1,\mathbf{P}_2,\dots,\mathbf{P}_R)$. In other words, PSHGCN can approximate any valid heterogeneous graph filter. Conversely, any valid heterogeneous graph filter can be expressed by PSHGCN, theoretically guaranteeing the effectiveness of PSHGCN.

### 5.2 Complexity

In Equation (6), $g$ is a $K$-order noncommutative polynomial. In theory, the number of terms in $g$ grows exponentially with the order $K$, i.e., $\frac{R^{K+1}-1}{R-1}$ terms. However, in real-world heterogeneous graphs, many types of nodes have no direct edges between them, e.g., such as authors and conferences in DBLP, which means that many terms $\hat{\mathbf{A}}_i\hat{\mathbf{A}}_j$ in $g$ are zero matrices. Therefore, we can ignore these terms in practice. Remarkably, these non-zero polynomial terms are analogous to the meta-paths commonly employed in most existing HGNNs [18, 28]. In other words, for heterogeneous graphs where all types of nodes are interconnected, the neighbors aggregated by these HGNNs also experience exponential growth with the length of the meta-paths. We can derive that the time complexity of PSHGCN in Equation (6) is $O(LKmd + nd^2)$, where $L$ denotes the number of non-zero terms in the polynomial $g$ with order $K$, $m$ denotes the maximum number of edges among $\{\hat{\mathbf{A}}_1,\hat{\mathbf{A}}_2,\dots,\hat{\mathbf{A}}_R\}$, $d$ is the feature dimension, and $O(nd^2)$ is the time complexity of the MLP. This complexity is expected to outperform many existing HGNNs, like HAN [28]. The difference is that PSHGCN doesn't need an attention mechanism for aggregation, and it learns the filter weights instead. In HAN, $L$ can be interpreted as the number of selected meta-paths, while $K$ can be seen as their maximum length.

For decoupled PSHGCN in Equation (7), its preprocessing time complexity is $O(LKmd)$, and the time complexity for training using mini-batch is $O(Bd^2)$, where $B$ denotes the batch size. This training complexity is significantly lower than that of SeHGNN [35], which is $O(BL^2d^2)$, where $L$ denotes the number of selected meta-paths. This reduction is primarily because SeHGNN needs to use a Transformer for feature fusion.

**Table 2: Node classification performance (Mean F1 scores ± standard errors) comparison of different methods on four datasets. Tabular results are presented in percentages, with the best result highlighted in bold and the runner-up underlined.**

| | DBLP | | ACM | | IMDB | | AMiner | |
|---|---|---|---|---|---|---|---|---|
| | Macro-F1 | Micro-F1 | Macro-F1 | Micro-F1 | Macro-F1 | Micro-F1 | Macro-F1 | Micro-F1 |
| GCN | $90.84_{\pm0.32}$ | $91.47_{\pm0.34}$ | $92.17_{\pm0.24}$ | $92.12_{\pm0.23}$ | $57.88_{\pm1.18}$ | $64.82_{\pm0.64}$ | $75.63_{\pm1.08}$ | $85.77_{\pm0.43}$ |
| GAT | $93.83_{\pm0.27}$ | $93.39_{\pm0.30}$ | $92.26_{\pm0.94}$ | $92.19_{\pm0.93}$ | $58.94_{\pm1.35}$ | $64.86_{\pm0.43}$ | $75.23_{\pm0.60}$ | $85.56_{\pm0.65}$ |
| GPRGNN | $91.66_{\pm1.01}$ | $92.45_{\pm0.76}$ | $92.36_{\pm0.28}$ | $92.28_{\pm0.27}$ | $58.90_{\pm1.15}$ | $64.84_{\pm0.81}$ | $75.32_{\pm0.67}$ | $86.13_{\pm0.58}$ |
| ChebNetII | $92.05_{\pm0.53}$ | $92.97_{\pm0.48}$ | $92.45_{\pm0.37}$ | $92.33_{\pm0.38}$ | $58.07_{\pm1.34}$ | $64.79_{\pm0.89}$ | $75.59_{\pm0.73}$ | $85.82_{\pm0.52}$ |
| RGCN | $91.52_{\pm0.50}$ | $92.07_{\pm0.50}$ | $91.55_{\pm0.74}$ | $91.41_{\pm0.75}$ | $58.85_{\pm0.26}$ | $62.05_{\pm0.15}$ | $63.03_{\pm2.27}$ | $82.79_{\pm1.12}$ |
| HAN | $91.67_{\pm0.49}$ | $92.05_{\pm0.62}$ | $90.89_{\pm0.43}$ | $90.79_{\pm0.43}$ | $57.74_{\pm0.96}$ | $64.63_{\pm0.58}$ | $63.86_{\pm2.15}$ | $82.95_{\pm1.33}$ |
| GTN | $93.52_{\pm0.55}$ | $93.97_{\pm0.54}$ | $91.31_{\pm0.70}$ | $91.20_{\pm0.71}$ | $60.47_{\pm0.98}$ | $65.14_{\pm0.45}$ | $72.39_{\pm1.79}$ | $84.74_{\pm1.24}$ |
| MAGNN | $93.28_{\pm0.51}$ | $93.76_{\pm0.45}$ | $90.88_{\pm0.64}$ | $90.77_{\pm0.65}$ | $56.49_{\pm3.20}$ | $64.67_{\pm1.67}$ | $71.56_{\pm1.63}$ | $83.48_{\pm1.37}$ |
| EMRGNN | $92.19_{\pm0.38}$ | $92.57_{\pm0.37}$ | $92.93_{\pm0.34}$ | $93.85_{\pm0.33}$ | $61.87_{\pm2.03}$ | $65.86_{\pm0.81}$ | $73.74_{\pm1.25}$ | $85.46_{\pm0.74}$ |
| MHGCN | $93.56_{\pm0.41}$ | $94.03_{\pm0.43}$ | $92.12_{\pm0.66}$ | $91.97_{\pm0.68}$ | $62.85_{\pm1.11}$ | $66.57_{\pm0.63}$ | $73.56_{\pm1.75}$ | $85.18_{\pm1.28}$ |
| SimpleHGN | $94.01_{\pm0.24}$ | $94.46_{\pm0.22}$ | $93.42_{\pm0.44}$ | $93.35_{\pm0.45}$ | $63.53_{\pm1.36}$ | $67.36_{\pm0.57}$ | $75.43_{\pm0.88}$ | $86.52_{\pm0.73}$ |
| HALO | $92.37_{\pm0.32}$ | $92.84_{\pm0.34}$ | $93.05_{\pm0.31}$ | $92.96_{\pm0.33}$ | $\underline{71.63_{\pm0.77}}$ | $\underline{73.81_{\pm0.72}}$ | $74.91_{\pm1.23}$ | $\underline{87.25_{\pm0.89}}$ |
| SeHGNN | $\underline{95.06_{\pm0.17}}$ | $\underline{95.42_{\pm0.17}}$ | $\underline{94.05_{\pm0.35}}$ | $\underline{93.98_{\pm0.36}}$ | $67.11_{\pm0.25}$ | $69.17_{\pm0.43}$ | $\underline{76.83_{\pm0.57}}$ | $86.96_{\pm0.64}$ |
| PSHGCN | $\mathbf{95.27_{\pm0.13}}$ | $\mathbf{95.61_{\pm0.12}}$ | $\mathbf{94.35_{\pm0.23}}$ | $\mathbf{94.27_{\pm0.23}}$ | $\mathbf{72.33_{\pm0.57}}$ | $\mathbf{74.46_{\pm0.32}}$ | $\mathbf{77.26_{\pm0.75}}$ | $\mathbf{88.21_{\pm0.31}}$ |

## 6 EXPERIMENTS

In this section, we conduct extensive experiments to assess the performance of PSHGCN against the state-of-the-art HGNNs for tasks involving node classification and link prediction. Furthermore, we evaluate the scalability of PSHGCN by employing the Open Graph Benchmark (OGB). Finally, we provide an in-depth model analysis from various perspectives. All the experiments are carried out on a machine with an NVIDIA Tesla A100 GPU (80 GB memory), Intel Xeon CPU (2.30 GHz) with 64 cores, and 512 GB of RAM.

### 6.1 Node Classification

**Datasets and Setting.** For the node classification task, we evaluate PSHGCN on four widely used heterogeneous graphs, including three academic citation heterogeneous graphs DBLP [18], ACM [18] and AMiner [29], and a movie rating graph IMDB [18]. Due to limited space, we provide dataset statistics in Table 6 within Appendix D.1 and offer a detailed introduction. For baselines, we first compare PSHGCN to four popular homogeneous GNNs, including GCN[15], GAT [27], GPRGNN [5] and ChebNetII [11], where GPRGNN and ChebNetII are two competitive spectral-based GNNs. Additionally, we compare PSHGCN to nine competitive HGNNs, including RGCN [20], HAN [28], GTN [39], MAGNN [7], EMRGNN [31], MHGCN [38], SimpleHGN [18], HALO [1] and Se-HGNN [35]. To ensure a fair comparison, we adopt the experimental setup used in the Heterogeneous Graph Benchmark (HGB) [18], and follow its standard split with the training/validation/test sets accounting for 24%/6%/70%. We use the existing baseline results provided by HGB [18]. For results that are not available, we use the officially released code and conduct a hyperparameter search based on the guidelines presented in their respective paper. For PSHGCN, we use Equation (6) as the propagation process and search the order $K$ from 1 to 5 in the polynomial $g$. We use a uniform distribution to

**Table 3: Link prediction performance (ROC-AUC/MRR ± standard errors). Results are presented in percent, with the best result highlighted in bold and the runner-up underlined.**

| | Amazon | | LastFM | |
|---|---|---|---|---|
| | ROC-AUC | MRR | ROC-AUC | MRR |
| GCN | $92.84_{\pm0.34}$ | $\underline{97.05_{\pm0.12}}$ | $59.17_{\pm0.31}$ | $79.38_{\pm0.65}$ |
| GAT | $91.65_{\pm0.80}$ | $96.58_{\pm0.26}$ | $58.56_{\pm0.66}$ | $77.04_{\pm2.11}$ |
| RGCN | $86.32_{\pm0.28}$ | $93.92_{\pm0.16}$ | $57.21_{\pm0.09}$ | $77.68_{\pm0.17}$ |
| GATNE | $77.39_{\pm0.50}$ | $92.04_{\pm0.36}$ | $66.87_{\pm0.16}$ | $85.93_{\pm0.63}$ |
| HetGNN | $77.74_{\pm0.24}$ | $91.79_{\pm0.03}$ | $62.09_{\pm0.01}$ | $83.56_{\pm0.14}$ |
| HGT | $88.26_{\pm2.06}$ | $93.87_{\pm0.65}$ | $54.99_{\pm0.28}$ | $74.96_{\pm1.46}$ |
| SeHGNN | $91.67_{\pm0.94}$ | $95.83_{\pm0.58}$ | $66.59_{\pm0.62}$ | $\underline{88.61_{\pm1.25}}$ |
| SimpleHGN | $\underline{93.40_{\pm0.62}}$ | $96.94_{\pm0.29}$ | $\underline{67.59_{\pm0.23}}$ | $\underline{90.81_{\pm0.32}}$ |
| PSHGCN | $\mathbf{94.12_{\pm0.58}}$ | $\mathbf{97.93_{\pm0.46}}$ | $\mathbf{69.25_{\pm0.63}}$ | $\mathbf{91.19_{\pm0.51}}$ |

randomly initialize the weights $w$ in $g$ and optimize them using gradient descent, consistent with spectral-based GNNs [5, 10, 11]. More details of hyper-parameters and settings are listed in Appnedix D.2.

**Results.** We use the mean F1 scores with standard errors over five runs as the evaluation metric. The results are presented in Table 2, with the top two performing results highlighted in bold and underlined, respectively. We first observe that the spectral-based GNNs, specially designed for homogeneous graphs, outperform certain HGNNs, like HAN. This suggests spectral-based GNNs' promising effectiveness even on heterogeneous graphs. Furthermore, PSHGCN outperforms other methods on all datasets, attributed to its capability of learning various valid heterogeneous graph filters. Notably, when compared to SeHGNN, PSHGCN achieves superior performance without relying on tricks like label propagation to enhance features. Instead, it directly learns the coefficients of the

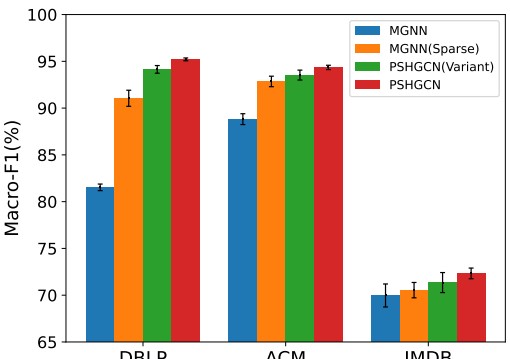 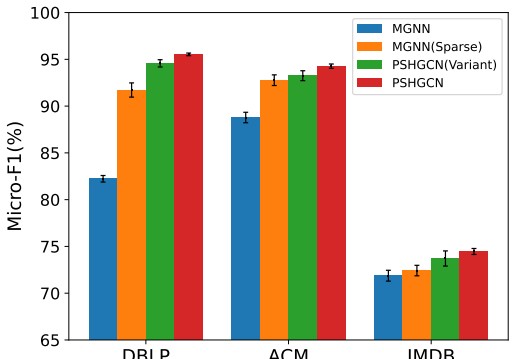

**Figure 2: Comparison of MGNN [2], PSHGCN, and its variant on node classification.**

polynomial $g$ in Equation (6). Nevertheless, PSHGCN outperforms SeHGNN on all datasets, with a significant advantage on IMDB and AMiner. These results highlight PSHGCN's effectiveness for heterogeneous graphs and its strong ability to learn filters.

## 6.2 Link Prediction

**Datasets and Setting.** For the link prediction task, we use two datasets Amazon and LastFM from HGB [18] to evaluate the performance of PSHGCN. We compare PSHGCN with eight methods, including two famous GNNs: GCN [15] and GAT [27], six competitive HGNNs: RCGN [20], GATNE [4], HetGNN [40], HGT [14], SeHGNN [35] and SimpleHGN [18]. We follow the experimental setup provided by HGB. The task of link prediction is cast as a binary classification problem, with the splitting of edges as follows: 81% for training, 9% for validation, and 10% for testing. Then the graph is reconstructed solely using the edges from the training set. We use randomly sampled 2-hop neighbors as the negative test set, as suggested by HGB. For baselines, we use the results provided in HGB, with the exception of SeHGNN. For SeHGNN, we use their publicly available code and conduct a hyperparameter search in accordance with the details in their paper. It's worth noting that due to the challenge of applying SeHGNN's label propagation technique directly to link prediction, we have excluded this component from our implementation. For PSHGCN, we employ the same implementation as utilized for the node classification task described in the previous subsection. We explore both dot product and DistMult [34] decoders, following the approach of SimpleHGN [18]. For more specific hyperparameter settings, please refer to Appendix D.3.

**Results.** We evaluate link prediction using mean ROC-AUC (area under the ROC curve) and MRR (mean reciprocal rank) with standard errors, over five repeated runs. The results are presented in Table 3. We observe that PSHGCN consistently outperforms other methods on both datasets. This underscores the effectiveness of PSHGCN in the link prediction task and demonstrates the feasibility of designing heterogeneous graph convolutions from the spectral domain. Notably, PSHGCN exhibits a significant advantage over SeHGNN, which we attribute to its enhanced expressiveness and flexibility. PSHGCN can derive diverse heterogeneous graph filters by directly learning coefficients, whereas SeHGNN relies on manually selected meta-paths and incorporates techniques like label propagation to boost its performance.

**Table 4: Node classification performance (Mean accuracies ± standard errors) on ogbn-mag, where the symbol "*" denotes the usage of extra embeddings and multi-stage training. The best results are highlighted in bold.**

| Methods | Validation accuracy | Test accuracy |
|---------|--------------------|--------------|
| RGCN | 48.35±0.36 | 47.37±0.48 |
| HGT | 49.89±0.47 | 49.27±0.61 |
| NARS | 51.85±0.08 | 50.88±0.12 |
| SAGN | 52.25±0.30 | 51.17±0.32 |
| GAMLP | 53.23±0.41 | 51.63±0.22 |
| SeHGNN | 55.95±0.11 | 53.99±0.18 |
| PSHGCN | **56.16±0.21** | **54.57±0.16** |
| SAGN* | 55.91±0.17 | 54.40±0.15 |
| GAMLP* | 57.02±0.41 | 55.90± 0.27 |
| SeHGNN* | 59.17±0.09 | 57.19±0.12 |
| PSHGCN* | **59.43±0.15** | **57.52±0.11** |

## 6.3 Scalability

To evaluate the scalability of PSHGCN, we conduct a node classification task on the large-scale heterogeneous graph ogbn-mag from the Open Graph Benchmark (OGB). We compare six baselines listed on the OGB leaderboard: RGCN [20], HGT [14], NARS [37], SAGN [24], GAMLP [42] and SeHGNN [35]. We use results on the leaderboard for these baselines. For PSHGCN, we use the decoupled version described in Equation (7), and more hyperparameter settings are listed in Appendix D.4.

Table 4 shows the mean accuracies with standard errors over five runs. We use the symbol * to denote the usage of extra embeddings (e.g., ComplEx embedding) and multi-stage training, which are commonly used in the baselines. We observe that PSHGCN has achieved a new SOTA result on ogbn-mag, underscoring the effectiveness and scalability of decoupled PSHGCN. Compared to the non-decoupled PSHGCN, the decoupled version relies more on the original node features since it does not utilize encoders like MLPs to transform the features before filtering. In fact, it is worth further exploration to investigate how to extend the non-decoupled PSHGCN to large-scale datasets using techniques such as sampling, and this also holds true for Spectral-based GNNs.

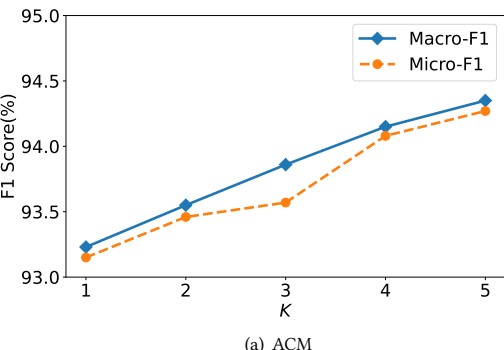

(a) ACM

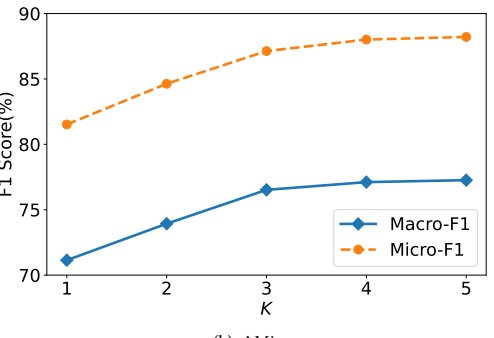

(b) AMiner

**Figure 3: Node classification performance of PSHGCN with respect to the order $K$.**

## 6.4 Model Analysis

**Impact of the positive semidefinite.** To investigate the impact of the positive semidefinite constraint, we compared PSHGCN and its variant without this constraint, as well as MGNN [2], the only method attempting to design heterogeneous graph convolution in the spectral domain. Specifically, MGNN uses noncommutative polynomials to create convolutions for multigraphs, which can be expressed as $\mathbf{XW}_0 + \sum_{k=1}^{K} \sum \hat{\mathbf{A}}_{r_1} \hat{\mathbf{A}}_{r_2} \cdots \hat{\mathbf{A}}_{r_k} \mathbf{XW}_{r_1, r_2, ..., r_k}$, where $\mathbf{W}$ are learnable weight matrices. These weight matrices implicitly define the polynomial coefficients. For the variant of PSHGCN without the positive semidefinite constraint, we achieve it by removing $g^\top$ in Equation (6). In the implementation, we used the code provided by the authors for MGNN. Unfortunately, this code stores the adjacency matrices in dense form, limiting the choice of higher-order $K$. So, we developed a sparse version based on the original code, denoted as MGNN (Sparse). For both MGNN (Sparse) and PSHGCN (Variant), our experiments followed the same settings as PSHGCN, with a search for polynomial orders ranging from 2 to 10. Figure 2 presents the results of our comparison. First, we observe that PSHGCN and its variants outperform MGNN and MGNN (Sparse), highlighting the effectiveness of directly learning polynomial coefficients, a finding consistent with research in spectral-based GNNs. Additionally, PSHGCN outperforms PSHGCN (Variant), especially with smaller standard errors over multiple repeated runs. This result underscores the significance of the positive semidefinite constraint in learning heterogeneous graph filters. It enhances learning ability and stability in practice while also ensuring the learned filters are always theoretically valid.

**Time comparison.** We conduct node classification on DBLP to evaluate the time cost (per epoch) and memory cost for several representative models, including GTN, MAGNN, HAN, MHGCN, EMRGNN, RGCN, HALO, SimpleHGN, SeHGNN, and PSHGCN. The results are shown in Figure 4. We found that PSHGCN is comparable to the advanced methods but significantly outperforms early methods such as HAN and RGCN. This is due to PSHGCN having a simple structure and no attention mechanism or other modules. In addition, we provide a comparison between decoupled PSHGCN and SeHGNN on ogbn-mag in Appendix D.5. As analyzed in Section 5.2, decoupled PSHGCN is more efficient than SeHGNN.

**Sensitivity of the order $K$.** We investigate the impact of the order $K$ in the polynomial $g$ on the performance of PSGCN. Figure 3 displays the node classification F1 scores with respect to the order

$K$ on ACM and AMiner (more results are listed in Appendix D.5). We find that the performance of PSHGCN increases gradually with increasing $K$, which is consistent with the theory of polynomial approximation in graph convolution.

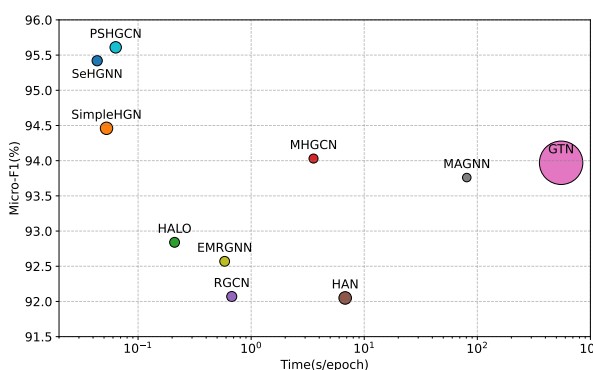

**Figure 4: Time and Memory Comparison for HGNNs on DBLP. The area of the circles corresponds to the (relative) memory consumption of the respective models.**

## 7 DISCUSSION & CONCLUSION

This paper introduces PSHGCN, a novel heterogeneous convolutional network that creates heterogeneous graph convolutions in the spectral domain. Through the utilization of positive noncommutative polynomials, PSHGCN enables effective learning of diverse valid heterogeneous graph filters. Experimental results demonstrate that PSHGCN achieves superior performance in node classification and link prediction tasks compared to existing methods. Notably, to our knowledge, this paper is the first attempt to obtain graph convolutions by directly learning the weights of spectral graph filters on heterogeneous graphs. Extensive experiments demonstrate the effectiveness of our proposed methods. Consequently, it opens up several directions for future research. (1) As mentioned in Section 6.3, it would be meaningful to explore alternative approaches, such as sampling or graph sparsification, to improve the scalability of PSHGCN. (2) Further investigation into the spectral analysis of PSHGCN makes sense. This involves defining the Fourier transform on heterogeneous graphs. Although some methods have been attempted with techniques like joint block diagonalization [2] or Jordan decomposition [21], these methods are not as intuitive or effective as those in homogeneous graphs.

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

# A NOTATIONS

**Table 5: Summation of main notations in this paper.**

| Notation | Description |
|---|---|
| $G = (V, E)$ | undirected homogeneous graph with node and edge sets $V$ and $E$ |
| $\hat{\mathbf{A}}, \mathbf{L}$ | the normalized adjacency and Laplacian matrix of graph $G = (V, E)$ |
| $h(\mathbf{L})$ | the spectral graph filter of graph $G = (V, E)$ |
| $G = (V, E, \phi, \psi)$ | heterogeneous graph with node and edge sets $V$ and $E$, node type $\phi$ and edge type $\psi$ |
| $n$ | the node number of graph $G = (V, E, \phi, \psi)$ |
| $R$ | the number of edge types of $G = (V, E, \phi, \psi)$ |
| $G_r$ | the sub-graph with only type of edge of graph $G = (V, E, \phi, \psi)$ |
| $\hat{\mathbf{A}}_r, \mathbf{L}_r$ | the normalized adjacency and Laplacian matrix of sub-graph $G_r$ |
| $\mathbf{P}_r$ | refers to either $\hat{\mathbf{A}}_r$ or $\mathbf{L}_r$ |
| $\mathbf{X}, \mathbf{x}$ | node feature matrix and graph signal vector |
| $h(\mathbf{P}_1, \mathbf{P}_2, \ldots, \mathbf{P}_r)$ | the spectral heterogeneous graph filter |

# B PROOF OF LEMMA 4.1

PROOF. We assume that the output of function $\gamma(\mathbf{P}_1, \mathbf{P}_2, \cdots, \mathbf{P}_R)$ is represented by the matrix $\mathbf{N}$. According to Lemma 5.1, $\mathbf{N}$ is a real symmetric positive semidefinite matrix. We perform an eigen-decomposition of $\mathbf{N}$ and express it as $\mathbf{N} = \mathbf{Q}\Sigma\mathbf{Q}^\top$, where $\mathbf{Q}$ is the matrix of eigenvectors and $\Sigma = \text{diag}[\sigma_1, \sigma_2, \cdots, \sigma_n]$ is the matrix of eigenvalues. Notably, the eigenvalues $\sigma_i$ satisfy $\sigma_i \geq 0$. Then, we have

$$\alpha \left[\alpha\mathbf{I} + (1-\alpha)\gamma(\mathbf{P}_1, \mathbf{P}_2, \cdots, \mathbf{P}_R)\right]^{-1} = \alpha\left[\alpha\mathbf{I} + (1-\alpha)\mathbf{N}\right]^{-1}.$$

We find that the matrix $\alpha\left[\alpha\mathbf{I} + (1-\alpha)\mathbf{N}\right]^{-1}$ is real and symmetric. Its eigenvalues are given by $\frac{\alpha}{\alpha+(1-\alpha)\sigma_i}$ for $i = 1, 2, \cdots, n$. It is evident that $\frac{\alpha}{\alpha+(1-\alpha)\sigma_i} > 0$ for $\alpha \in (0, 1)$ and $\sigma_i \geq 0$. Consequently, the matrix $\alpha\left[\alpha\mathbf{I} + (1-\alpha)\mathbf{N}\right]^{-1}$ is a real positive semidefinite matrix. □

# C PSEUDOCODE

Algorithm 1 presents the pseudocode for PSHGCN. In this context, $\{\mathbf{X}^{\phi_i}|i = 1, 2, \ldots, |\mathcal{T}v|\}$ represents a collection of node feature matrices of different types. For example, $\mathbf{X}^{\phi_i}$ corresponds to the feature matrix of a node with node type $\phi_i$. The dimension of the feature matrix $\mathbf{X}^{\phi_i}$ is $|\phi_i| \times d_{\phi_i}$, and following the concatenation operation in step 5, $\mathbf{H}$ will have dimensions of $n \times d$, where $d$ is the hidden dimension.

Algorithm 2 presents the pseudocode for decoupled PSHGCN. In step 16, $c_{r_1, r_2, \ldots, r_k}$ are the coefficients of the $\hat{\mathbf{A}}_{r_1}\hat{\mathbf{A}}_{r_2}\cdots\hat{\mathbf{A}}_{r_k}$ term in the expansion of $g(\hat{\mathbf{A}}_1, \hat{\mathbf{A}}_2, \ldots, \hat{\mathbf{A}}_R)^\top g(\hat{\mathbf{A}}_1, \hat{\mathbf{A}}_2, \ldots, \hat{\mathbf{A}}_R)$ and $c_0 = w_0^2$.

---

**Algorithm 1:** Pseudocode of PSHGCN

**Input:** heterogeneous graph $G = (V, E, \phi, \psi)$, raw node feature matrices $\{\mathbf{X}^{\phi_i}|i = 1, 2, \ldots, |\mathcal{T}_v|\}$, order $K$.

**Parameter:** polynomial coefficients $w_0$ and $w_{r_1, r_2, \ldots, r_k}$, $\text{MLP}_{in}^{\phi_i}$ for feature projection, $\text{MLP}_{out}$ for downstream task.

**Output:** The node embedding $\mathbf{Z}$ of graph $G$.

1 Get the normalized adjacency matrices $\{\hat{\mathbf{A}}_r|r = 1, 2, \ldots, R\}$;
2 Randomly initialize coefficients $w_0$ and $w_{r_1, r_2, \ldots, r_k}$;
3 **for** $i = 1$ to $|\mathcal{T}_v|$ **do**
4 $\quad \mathbf{H}^{\phi_i} \leftarrow \text{MLP}_{in}^{\phi_i}(\mathbf{X}^{\phi_i})$;
5 $\mathbf{H} \leftarrow \text{concatenate}\left(\{\mathbf{H}^{\phi_i}|i = 1, 2, \ldots, |\mathcal{T}_v|\}\right)$;
6 $\mathbf{Y}' \leftarrow \left(w_0\mathbf{I} + \sum_{k=1}^{K}\sum w_{r_1, r_2, \ldots, r_k}\left(\hat{\mathbf{A}}_{r_1}\hat{\mathbf{A}}_{r_2}\cdots\hat{\mathbf{A}}_{r_k}\right)\right)\mathbf{H}$;
7 $\mathbf{Y} \leftarrow \left(w_0\mathbf{I} + \sum_{k=1}^{K}\sum w_{r_1, r_2, \ldots, r_k}\left(\hat{\mathbf{A}}_{r_1}\hat{\mathbf{A}}_{r_2}\cdots\hat{\mathbf{A}}_{r_k}\right)\right)^\top\mathbf{Y}'$;
8 $\mathbf{Z} \leftarrow \text{MLP}_{out}(\mathbf{Y})$;
9 **return** $\mathbf{Z}$;

---

**Algorithm 2:** Pseudocode of decoupled PSHGCN

**Input:** heterogeneous graph $G = (V, E, \phi, \psi)$, raw node feature matrices $\{\mathbf{X}^{\phi_i}|i = 1, 2, \ldots, |\mathcal{T}_v|\}$, order $K$.

**Parameter:** polynomial coefficients $w_0$ and $w_{r_1, r_2, \ldots, r_k}$, $\mathbf{W}^{\phi_i}$ for feature transformation, $\text{MLP}_{out}$ for downstream task.

**Output:** The node embedding $\mathbf{Z}$ of graph $G$.

1 Get the normalized adjacency matrices $\{\hat{\mathbf{A}}_r|r = 1, 2, \ldots, R\}$;
2 Randomly initialize coefficients $w_0$ and $w_{r_1, r_2, \ldots, r_k}$;
3 % Preprocessing
4 **for** $i = 1$ to $|\mathcal{T}_v|$ **do**
5 $\quad \tilde{\mathbf{X}}^{\phi_i} \leftarrow$ Convert the dimension of $\mathbf{X}^{\phi_i}$ to $n \times d_{\phi_i}$;
6 **for** $i = 1$ to $|\mathcal{T}_v|$ **do**
7 $\quad \mathbf{H}_0^{\phi_i} \leftarrow \tilde{\mathbf{X}}^{\phi_i}$;
8 $\quad$ **for** each $r_1, r_2, \ldots, r_k$ **do**
9 $\quad\quad \mathbf{H}_{r_1, r_2, \ldots, r_k}^{\phi_i} \leftarrow \hat{\mathbf{A}}_{r_1}\hat{\mathbf{A}}_{r_2}\cdots\hat{\mathbf{A}}_{r_k}\tilde{\mathbf{X}}^{\phi_i}$;
10 % Training
11 **for** $i = 1$ to $|\mathcal{T}_v|$ **do**
12 $\quad \tilde{\mathbf{H}}_0^{\phi_i} \leftarrow \mathbf{H}_0^{\phi_i}\mathbf{W}^{\phi_i}$;
13 $\quad \tilde{\mathbf{H}}_{r_1, r_2, \ldots, r_k}^{\phi_i} \leftarrow \mathbf{H}_{r_1, r_2, \ldots, r_k}^{\phi_i}\mathbf{W}^{\phi_i}$;
14 $\mathbf{H}_0 \leftarrow \sum_{i=1}^{|\mathcal{T}_v|}\tilde{\mathbf{H}}_0^{\phi_i}$;
15 $\mathbf{H}_{r_1, r_2, \ldots, r_k} \leftarrow \sum_{i=1}^{|\mathcal{T}_v|}\tilde{\mathbf{H}}_{r_1, r_2, \ldots, r_k}^{\phi_i}$;
16 $\mathbf{Y} = c_0\mathbf{H}_0 + \sum_{k=1}^{K}\sum c_{r_1, r_2, \ldots, r_k}\mathbf{H}_{r_1, r_2, \ldots, r_k}$;
17 $\mathbf{Z} = \text{MLP}_{out}(\mathbf{Y})$;
18 **return** $\mathbf{Z}$;

**Table 6: Statistics of datasets on the node classification task.**

| Dataset | #Nodes | #Node Types | #Edges | #Edges Types | Target | #Classes |
|---|---|---|---|---|---|---|
| DBLP | 26,128 | 4 | 239,566 | 6 | author | 4 |
| ACM | 10,942 | 4 | 547,872 | 8 | paper | 3 |
| IMDB | 21,420 | 4 | 86,642 | 6 | movie | 5 |
| AMiner | 55,783 | 3 | 153,676 | 4 | paper | 4 |
| Ogbn-mag | 1,939,743 | 4 | 21,111,007 | 4 | paper | 349 |

**Table 7: Statistics of datasets on the link prediction task.**

| Dataset | #Nodes | #Node Types | #Edges | #Edges Types | Target |
|---|---|---|---|---|---|
| Amazon | 10,099 | 1 | 148,659 | 2 | product-product |
| LastFM | 20,612 | 3 | 141,521 | 3 | user-artist |

# D ADDITIONAL EXPERIMENTAL DETAILS

## D.1 Datasets and Baselines

**Datasets.** We use five common real-world heterogeneous datasets for node classification, including three academic citation heterogeneous graphs: DBLP [18], ACM [18], and AMiner [29], as well as a heterogeneous graph based on movie ratings, IMDB [18], and a large-scale academic citation heterogeneous graph known as ogbn-mag [13]. The statistics for these datasets can be found in Table 6. Additionally, we utilize two prevalent real-world heterogeneous datasets for link prediction: one stemming from product purchase data, Amazon [18], and the other from online music data, LastFM [18]. You can refer to Table 7 for the statistics of these two datasets. Further details about each of these datasets are provided below.

- **DBLP** is a computer science bibliography website that contains papers published between 1994 and 2014 from 20 conferences across four research fields. The dataset comprises four types of nodes: authors (A), papers (P), terms (T), and venues (V), as well as six types of edges: A-P, P-A, P-V, V-P, P-T, and T-P. For meta-path-based HGNNs, the utilized meta-paths are APA, APTPA, and APVPA.
- **ACM** is an academic citation network that encompasses papers from three classes: Database, Wireless Communication, and Data Mining. The dataset consists of four types of nodes: authors (A), papers (P), subjects (S), and fields (F), along with eight types of edges: A-P, P-A, P-c-P, P-r-P, P-S, S-P, P-K, and K-P (where 'c' denotes citation relation and 'r' denotes reference relation). For meta-path-based HGNNs, the used meta-paths are PAP, PSP, PcPAP, PcPSP, PrPAP, and PrPSP.
- **IMDB** is an online platform that provides information about movies and their associated details. The movies are categorized into five classes: action, comedy, drama, romance, and thriller. The dataset encompasses four types of nodes: movies (M), directors (D), actors (A), and keywords (K), and includes six types of edges: M-A, A-M, M-D, D-M, M-K, and K-M. For meta-path-based HGNNs, the utilized meta-paths include MDM, MAM, DMD, DMAMD, AMA, and AMDMA.
- **AMiner** is also an academic citation network that includes four types of papers. The dataset includes three types of nodes: authors (A), papers (P), and references (R), with four types of edges: A-P, P-A, R-P, and P-R. For meta-path-based HGNNs, the used meta-paths are PAP and PRP.
- **Ogbn-mag** is a large-scale heterogeneous network derived from a subset of the Microsoft Academic Graph. It includes types

of nodes: papers (P), authors (A), institutions (I), and fields of study (F), along with four types of directed relations. For more detailed information, please refer to the Open Graph Benchmark (OGB) [13].

- **Amazon** is an online retail platform containing a vast array of electronic products within its network, interconnected by co-viewing and co-purchasing links. The dataset consists of a single node type, products (P), accompanied by two distinct types of edges: viewing and purchasing.
- **LastFM** is an online music website. The dataset comprises three node categories: users (U), artists (A), and tags (T), interconnected by three types of edges: U-U, U-A, and A-T. For meta-path-based HGNNs, the employed meta-paths encompass UU, UAU, UATAU, AUA, ATA, and AUUA.

**Baseline Implementations.** For GCN, GAT, RGCN, HAN, GTN, MAGNN, GATNE, HetGNN, HGT, and SimpleHGN, we use the Heterogeneous Graph Benchmark (HGB) implementations [18]. For other baselines, we use the implementation released by the authors.

- **HGB:** https://github.com/THUDM/HGB
- **GPR-GNN:** https://github.com/jianhao2016/GPRGNN
- **ChebNetII:** https://github.com/ivam-he/ChebNetII
- **EMRGNN:** https://github.com/tuzibupt/EMR
- **MHGCN:** https://github.com/NSSSJSS/MHGCN
- **HALO:** https://github.com/hongjoon0805/HALO
- **SeHGNN:** https://github.com/ICT-GIMLab/SeHGNN
- **MGNN:** https://github.com/landonbutler/MultigraphNN

## D.2 Node classification in Section 6.1

We follow the experimental setup provided by the Heterogeneous Graph Benchmark (HGB) [18] and utilize the baseline results already available in their paper. In cases where baseline results are not accessible, we rely on the officially released code and perform a hyperparameter search following the guidelines outlined in the respective paper.

For our PSHGCN model, we first apply a feature projection layer to align node features, ensuring that different types of nodes share the same dimensional feature space. This feature projection layer is commonly employed by various popular HGNNs [1, 18, 31, 35]. For the MLPs in PSHGCN, we search the hidden dimension from the set $\{32, 64, 128, 256\}$. Similarly to many popular spectral-based GNNs [10, 11, 30], we train the linear and convolutional layers using distinct learning rates and weight decays. Specifically, we employ $\text{lr}_{\text{mlp}}$ and $L_{2_{\text{mlp}}}$ to represent the learning rate and weight decay for the linear layers, while $\text{lr}_{\text{conv}}$ and $L_{2_{\text{conv}}}$ are used for the convolutional layers. The hyperparameters of PSHGCN for the node classification task are presented in Table 8.

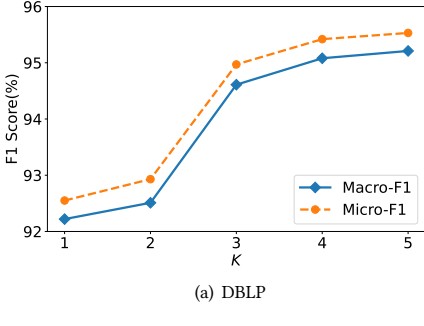

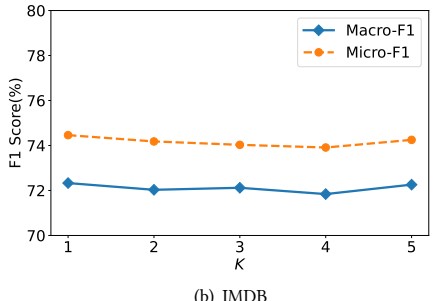

(a) DBLP

(b) IMDB

**Figure 5: Node classification performance of PSHGCN with respect to the order $K$.**

**Table 8: The hyper-parameters of PSHGCN for node classification in Section 6.1.**

| Dataset | hidden | $K$ | dropout | $\text{lr}_{\text{mlp}}$ | $L2_{\text{mlp}}$ | $\text{lr}_{\text{conv}}$ | $L2_{\text{conv}}$ |
|---------|--------|-----|---------|------|------|------|------|
| DBLP | 64 | 5 | 0.10 | 0.006 | 0.0 | 0.002 | 0.8 |
| ACM | 256 | 5 | 0.25 | 0.004 | 0.0 | 0.004 | 0.8 |
| IMDB | 256 | 1 | 0.70 | 0.0005 | 5e-4 | 0.008 | 0.0 |
| AMiner | 32 | 5 | 0.35 | 0.008 | 5e-4 | 0.008 | 0.3 |

**Table 9: Comparison of PSHGCN and SeHGNN on ogbn-mag.**

| Method | Accuracy (%) | #Params | Time (s/epoch) |
|--------|-------------|---------|----------------|
| SeHGNN | 57.19±0.12 | 8,371,231 | 7.8218 |
| PSHGCN | 57.52±0.11 | 4,852,434 | 5.8989 |

## D.3  Link prediction in Section 6.2

For the link prediction task, we follow the experimental setup provided by HGB. The task of link prediction is cast as a binary classification problem, with the splitting of edges as follows: 81% for training, 9% for validation, and 10% for testing. Then the graph is reconstructed solely using the edges from the training set. For PSHGCN, we use the same implementation as used for the node classification task. we search the hidden dimension of MLPs from the set {32, 64, 128, 256}, learn rating from the set {0.0005, 0.001, 0.005, 0.01, 0.05}, weight decays from the set {0.0, 4e-5, 3e-5, 0.001, 0.05, 0.1, 0.5 }, and dropout from {0.1, 0.2, 0.5,0.8}.

## D.4  Node classification on ogbn-mag

For the larger-scale dataset ogbn-mag, we use the leaderboard results provided by the Open Graph Benchmark (OGB)[13] for the baselines. Regarding PSHGCN, we set the value of $K$ to 4 in Equation(7) and perform the preprocessing step to calculate $\hat{\mathbf{A}}_{r_1}, \hat{\mathbf{A}}_{r_2}, \cdots, \hat{\mathbf{A}}_{r_k}\mathbf{X}$. In cases where certain node types lack raw features, we initialize their features randomly. As for PSHGCN*, we employ the ComplEx algorithm [26] to generate additional embeddings and adopt multi-stage learning. In the multi-stage learning process, we select test nodes with confident predictions in the last training stage, incorporate them into the training set, and retrain the model in a new stage [35, 42]. Since the most advanced methods [35, 42] on ogbn-mag currently utilize label propagation to enhance training, we also include the label propagation module. Regarding the hyperparameters, the hidden dimension is set to 1024, the dropout rate is 0.5, the learning rate is 0.001, and the weight decay is 0.0. Further implementation details are available in the code repository.

## D.5  More Experimental Results

Table 9 shows the comparison between decoupled PSHGCN and SeHGNN on ogbn-mag. As analyzed in Section 5.2, decoupled PSHGCN is more efficient than SeHGNN. Figure 5 displays the node classification F1 scores with respect to the order $K$ on DBLP and IMDB datasets. Notably, we observe either a gradual improvement or stabilization in the performance of PSHGCN as $K$ increases, aligning with the findings discussed in the ablation study Section 6.4.

