# OpenReview forum: "Spectral Heterogeneous Graph Convolutions via Positive Noncommutative Polynomials"
_ACM.org/TheWebConf/2024/Conference — TheWebConf24 Oral_

### Official Review · Reviewer_gXv8 · 2023-11-17

**Novelty:** 5
**Technical Quality:** 6

**Review:**

**Summary**:

In this paper, the authors propose a spectral domain-based Heterogeneous Graph Neural Networks (HGNNs) methods PSHGCN with a positive spectral heterogeneous graph convolution. PSHGCN enables heterogeneous graph filter to satisfy the requirement of a positive semidefinite constraint. Their experimental results show that PSHGCN can learn diverse heterogeneous graph filters and outperform all baselines on open benchmarks.

&nbsp;

**Strengths**:

**S1**. The paper thoroughly explores the design philosophy of PSHGCN from multiple perspectives. Specifically, it delves into detailed comparisons with existing spectral HGNNs, examining the graph optimization and complexity perspectives.

**S2**. The paper is well-written and easy to follow.

**S3**. Code is provided.

&nbsp;

**Weaknesses**:

**W1**. No significant test between the best performance and the second-best performance in Table 2. The difference between the best-performing and the second-best methods is slight in all datasets. A significant test should be needed.

**Questions:**

**Q1**. Please further discuss why models that are not equivalent to this noncommutative polynomial would lead to limits in their expressiveness.

**Q2**. Can you provide statistical analyses, such as significant tests, of repeated trials to explain that the better result of PSHGCN is not accidental?

**Ethics Review Description:**

Not Applicable. The paper does not have any ethical considerations to address.

**Reviewer Confidence:**

3: The reviewer is confident but not certain that the evaluation is correct

**Scope:**

4: The work is relevant to the Web and to the track, and is of broad interest to the community

---

### Official Review · Reviewer_Xyka · 2023-11-22

**Novelty:** 5
**Technical Quality:** 5

**Review:**

This paper investigates the field of heterogeneous graph neural networks (HGNNs). To enhance the expressive power, this paper presents a positive spectral heterogeneous graph convolution via polynomials and proposes PSHGCN.


Strength:

1. This paper presents an interesting idea to extend spectral graph filters to heterogeneous graphs.
2. Explicit time complexity is analyzed for the proposed.
3. The proposed PSHGCN is extensively evaluated in the experiments in terms of both efficiency and effectiveness.
4. Necessary theoretical analysis is provided.



Weakness:

1. It seems that the number of trainable weights in heterogeneous graph filters is exponential to the number of relations in heterogeneous graphs, which raises concerns about its scalability.
2. The total running times are missing for the tested large datasets. I am curious about the training time corresponding to the results in Table 4 for PSHGCN.



Minor issue:
PSHGCN should be defined before it is first used.

**Questions:**

Please see my weakness.

**Reviewer Confidence:**

3: The reviewer is confident but not certain that the evaluation is correct

**Scope:**

3: The work is somewhat relevant to the Web and to the track, and is of narrow interest to a sub-community

---

### Official Review · Reviewer_zcQH · 2023-11-24

**Novelty:** 4
**Technical Quality:** 5

**Review:**

This paper introduces a positive spectrum heterogeneous graph convolution based on positive semidefinite polynomials. Building upon this, a novel heterogeneous graph convolutional network, PSHGCN, is proposed. The fundamental principles of PSHGCN are demonstrated within a graph optimization framework.
Strengths:
1) This paper offers some theoretical analysis.
2) The proposed PSHGNN addresses the issues of poor theoretical guarantees and limited expressiveness observed in existing HGNNs.
3) Some experiments were conducted to demonstrate the effectiveness of the proposed method.
Weaknesses:
1) The issues addressed by the proposed PSHGCN in this paper are not novel problems, and there are some aspects of the specific implementation process that draw inspiration from previous works, indicating a lack of novelty.
2) Figure 1 lacks detailed explanations.
3) The paper mentions ‘simplifying the Sum of Squares form by utilizing a single polynomial’ in Section 4.3, but there is no theoretical proof provided for the feasibility of doing so.
4) The improvement in performance for node classification is not significant and the experiments lack comparisons with the latest methods.

**Questions:**

--How is the search for hyperparameters conducted?
--Why does GCN outperform many newer methods on the link prediction task in the Amazon dataset?

**Reviewer Confidence:**

3: The reviewer is confident but not certain that the evaluation is correct

**Scope:**

3: The work is somewhat relevant to the Web and to the track, and is of narrow interest to a sub-community

---

### Official Review · Reviewer_umC3 · 2023-11-24

**Novelty:** 4
**Technical Quality:** 4

**Review:**

This work aims to develop a heterogeneous graph neural network model from the spectral perspective that can learn arbitrary valid heterogeneous graph filters. The authors propose the Positive Spectral Heterogeneous Graph Convolutional Network (PSHGCN), which leverages spectral graph convolutions and positive noncommutative polynomials. This method ensures that the acquired graph filters are positive semidefinite, and its rationale is justified by a generalized graph optimization framework (also presented by the authors). Experiments examine PSHGCN's performance on both node classification and link prediction tasks, and ablations studies are given to justify its scalability and the influence of hyper-parameter K (the max order) on the model performance.


Pros:

1. The paper is well-organized and the structure of this paper is easy to follow.
2. Code is provided.
3. The experimental setting is very clear, details are provided in the main paper and supplementary.
4. Developing HGNN models from a spectral perspective is a meaningful topic.


Cons:

1. The performance improvement is marginal except on the IMDB dataset.
2. The model design seems to be the same as considering all possible metapaths within hop-k and learning the coefficient on each of the metapaths.
3. What would be the learned coefficient is not clearly presented.


My major concerns are in the model design (cons 2,3) and in the performance (cons 1).  In addition, I have a few questions listed in the Questions section. Therefore, I currently would like to vote for a reject.

**Questions:**

1. For coefficients cr1...rk in formula (7), are they learnable? Comparing (6) and (7), the only difference is we remove the first MLP step so that we can pre-compute the Y?
2. It seems to me that, g() is essentially considering all the possible combinatorial of adjacency matrix Ari within order k (i.e. considering all potential metapaths within hop-k). In that case, I am wondering, are all of these metapaths important and worth to be considered? Will too many metapaths bring any noise?
3. What kind of coefficients (i.e. cis in formula (7)) can the model learn? Can the author please provide some visualizations (e.g. with heatmaps) on this?
4. Are the reported results obtained with a non-decoupled version or a decoupled version? Can the author please provide both?

**Reviewer Confidence:**

3: The reviewer is confident but not certain that the evaluation is correct

**Scope:**

3: The work is somewhat relevant to the Web and to the track, and is of narrow interest to a sub-community

---

### Official Review · Reviewer_ic7Q · 2023-11-27

**Novelty:** 5
**Technical Quality:** 6

**Review:**

Summary: This paper designs convolution matrices for graph neural networks in the case where graphs have nodes and edges with different "types".  The convolution matrices are guaranteed to be positive semidefinite by virtue of the fact that they can be written in a sum of squares form.  The authors motivate this by recalling the connection between spectral graph filters and solutions to a certain energy functional minimization problem on graph signals.  In order for this connection to hold, the filter matrix must be positive semidefinite.

Pros:

1.) The development of the ideas is fairly clear.

2.) The empirical performance of the resulting methods at least matches baselines.

Cons:

1.) The empirical improvements are minimal, given the magnitudes of the reported variances.  In fact, this holds even in comparison to versions of the proposed architecture that do not enforce the positive semidefiniteness constraint.

2.) The theoretical contributions seem overstated.  Specifically, I am not sure if the expressiveness guarantee on page 5 holds when, as the authors do, one only uses a single monomial $g$ (as stated at the beginning of Section 4.3).

3.) The paper's approach is motivated by the fact that, if a filter $h$ is PSD, then it corresponds to the unique solution of equation (3) for some choice of the energy function $\gamma(L)$.  It is not clearly argued why this is so important for performance, though.

4.) Some notation needs to be reworked.  For example, in equation (3), the right-hand side seems to be the definition of the function $f$.  One should not write an optimization in this way.  One needs $\min_{y}$ on both sides.

**Questions:**

1.) Can the authors clarify points 2 and 3 in the "cons" section of this review?  Cons 1-3 are crucial in my evaluation of the paper.

2.) Can the authors define "meta-path" more precisely?

**Reviewer Confidence:**

3: The reviewer is confident but not certain that the evaluation is correct

**Scope:**

4: The work is relevant to the Web and to the track, and is of broad interest to the community

---

### Decision · Program_Chairs · 2024-01-22

**Decision:**

Accept (Oral)

**Comment:**

This paper develops a new heterogeneous graph method that applies the spectral method on a heterogeneous graph.

 Pros:
 * The paper applies a spectral method on heterogeneous graphs that commonly exist in the real-world setting. This is quite novel and is important for the field.
 * The paper also analyzes the computation complexity of the work and demonstrates its scalability both theoretically and empirically.
 * The method requires less configuration compared with the current SOTA while achieving performance comparable to or better than the current SOTA.

 Cons:
 * Although the authors have explained in the rebuttal about the model performance, the performance improvement is still quite minor except the IMDB dataset when comparing the strong baseline (SeHGNN)